# The Blood–Brain Barrier Is Unaffected in the *Ndufs4^−/−^* Mouse Model of Leigh Syndrome

**DOI:** 10.3390/ijms25094828

**Published:** 2024-04-29

**Authors:** Robin Reynaud-Dulaurier, Romain Clément, Sara Yjjou, Cassandra Cresson, Yasmina Saoudi, Mathilde Faideau, Michael Decressac

**Affiliations:** Inserm, U1216, CHU Grenoble Alpes, Grenoble Institut Neurosciences, Université Grenoble Alpes, 38000 Grenoble, France; robin.reynaud-dulaurier@univ-grenoble-alpes.fr (R.R.-D.); clementrom@outlook.fr (R.C.); sara.yjjou@univ-grenoble-alpes.fr (S.Y.); cassandra.cresson@univ-grenoble-alpes.fr (C.C.); yasmina.saoudi@univ-grenoble-alpes.fr (Y.S.); mathilde.decressac@univ-grenoble-alpes.fr (M.F.)

**Keywords:** blood–brain barrier, mitochondrial diseases, Leigh syndrome, AAV vector

## Abstract

Mitochondrial dysfunction plays a major role in physiological aging and in many pathological conditions. Yet, no study has explored the consequence of primary mitochondrial deficiency on the blood–brain barrier (BBB) structure and function. Addressing this question has major implications for pharmacological and genetic strategies aimed at ameliorating the neurological symptoms that are often predominant in patients suffering from these conditions. In this study, we examined the permeability of the BBB in the *Ndufs4^−/−^* mouse model of Leigh syndrome (LS). Our results indicated that the structural and functional integrity of the BBB was preserved in this severe model of mitochondrial disease. Our findings suggests that pharmacological or gene therapy strategies targeting the central nervous system in this mouse model and possibly other models of mitochondrial dysfunction require the use of specific tools to bypass the BBB. In addition, they raise the need for testing the integrity of the BBB in complementary in vivo models.

## 1. Introduction

The blood–brain barrier (BBB) is a highly selective and semi-permeable membrane delineated by endothelial cells of the brain vessels establishing an interface between vascular milieu and the central nervous system. Endothelial cells are supported by pericytes, both surrounded by a continuous, non-fenestrated basal lamina and astrocytic end-feet. Complex tight junctions between endothelial cells restrict the paracellular transport and force the molecules to take a transcellular route. Except small gaseous molecules or small lipophilic agents, the transcellular traffic requires the presence of specific transport systems, allowing the entry of nutrients, ions or neurotransmitters. Exchanges are highly regulated in order to maintain a brain microenvironment appropriate for proper neuronal function. It preserves the immune-privileged status of the brain and prevents the entry of blood-circulating neurotoxic agents [1].

The disruption of the BBB usually results in a higher blood–brain vessel permeability and consequently an increased vulnerability of the brain parenchyma to toxic endogenous or exogenous molecules. BBB disruption has been reported during physiological aging and in age-related diseases such as neurodegenerative disorders or diabetes [2,3,4]. Moreover, recent studies examined the integrity of the BBB in genetic disorders and found a breakdown of this barrier for instance in Huntington’s disease, X-linked adrenoleukodystrophy, cerebellar ataxia or genetically induced schizophrenia [5,6,7,8].

Mitochondrial dysfunction is a major hallmark of physiological senescence and is also documented in a large number of genetic and non-genetic diseases. Mitochondria are responsible for energy metabolism through oxidative phosphorylation involving the respiratory chain complexes. Studies have shown that the pharmacological inhibition of mitochondrial complex I or III activity in endothelial cells is responsible for an increase in BBB permeability both in vitro and in vivo [9,10]. In pathological conditions, it has been demonstrated that mitochondrial dysfunction in endothelial cells was, at least partly, responsible for BBB failure during ischemic stroke or septic encephalopathy [10,11]. In the context of primary mitochondrial impairment, coenzyme Q10 deficiency caused a leakage in an in vitro model of BBB that was associated with a disruption of tight junctions [12]. Surprisingly, the structural and functional status of the BBB has never been assessed in a mouse model of inherited mitochondrial disease.

Besides its pathological consequence, the disruption of the BBB can also be considered as a therapeutic opportunity. It has been estimated that more than 90% of small therapeutic molecules are not able to cross the BBB [13]. This include viral vectors used in gene replacement strategies for brain disorders [14]. Thus, an increased permeability of the BBB can be seen as an advantage as it facilitates the entry of therapeutic agents inside the cerebral parenchyma. For instance, in the case of Leigh syndrome (LS), one of the most common and severe mitochondrial diseases, pre-clinical studies testing small molecules showed remarkable therapeutic effects in *Ndufs4^−/−^* mice [15,16,17,18]. However, conflicting results exist in the literature regarding the ability of theses compound to cross the BBB. Thus, one can wonder whether their beneficial effects were facilitated by an increase in BBB permeability or by an indirect action. In the same animal model of LS, gene replacement strategies to reinstate *Ndufs4* gene expression have been explored [19,20,21,22,23]. They used different approaches to bypass the BBB but none of them checked if the permeability of the BBB to adeno-associated viral (AAV) vectors was modified. Depending on this information, methods to deliver therapeutic agents could be adapted in this model as well as in other mitochondrial diseases.

In this study, we chose to evaluate the integrity of the BBB in the most commonly studied mouse model of LS, namely the *Ndufs4^−/−^* mouse [24,25]. *Ndufs4* encodes for a 17-kDa subunit of mitochondrial complex I. This mouse model is of particular interest because it replicates cardinal features of the human pathology. In particular, the neurological phenotype is severe and to the animals’ death around post-natal day 50. Some pathological hallmarks such as inflammation or lipid droplet accumulation could favor alterations of the BBB [14,26,27]. We first assessed the structural and molecular integrity of the BBB in one-month-old *Ndufs4^−/−^* mice. Then, we evaluated its permeability to an AAV9 vector as well as to exogenous and endogenous molecules of various sizes. Our results showed that the BBB of *Ndufs4^−/−^* mice was not impaired. This study confirms the necessity to use strategies promoting the passage of therapeutic agents across the BBB in this mitochondrial disease.

## 2. Results

### 2.1. Biochemical and Anatomical Analysis of the BBB in Ndufs4^−/−^ and Control Mice

Previous studies showed that the constitutive deletion of the *Ndufs4* gene in mice leads to a reduction in mitochondrial complex I activity that varies between tissues [20]. To assess the impact of Ndufs4 deletion on the BBB, we isolated brain vessels from 1-month-old control and *Ndufs4^−/−^* mice and we measured NADH consumption from the mitochondrial fraction (Figure 1A). Brain vessels from this mouse model of LS had a reduced mitochondrial complex I activity compared to the control group (Figure 1B) (*p* < 0.001). These results suggest that this genetic mouse model represents a relevant system on which to study the effect of mitochondrial dysfunction on BBB permeability.

At the level of tight junctions, several membrane-associated and scaffolding proteins such as occluding and claudin-5 participate in the closure of the BBB, thereby limiting paracellular passage. We analyzed the expression of these proteins by Western blot and found no difference between vessels isolated from control and *Ndufs4^−/−^* mice (Figure 1C) (Ps > 0.05). The expression of CD31/PECAM, a marker of adherent junction was not altered in *Ndufs4^−/−^* mice (*p* > 0.05). In addition, we verified the abundance of astrocytic end-feet, another component of the BBB, by detection of the marker GFAP and we showed no difference between the two genotypes (Figure 1C) (*p* > 0.05).

We then studied the morphology of blood vessels in 1-month-old control and *Ndufs4^−/−^* mice. At this age, mice already exhibit pathological defects in several organs [24,27]. This is also a timepoint at which we and others demonstrated that the pathological phenotype can be rescued by a single administration of a brain-penetrating vector [22,23,27]. To examine the architecture of major cerebral vessels, mice received an intravenous injection of Evans Blue and brains were processed for tissue clearing followed by light sheet fluorescence microscopy (Figure 1E). Three-dimensional reconstructions showed that the brain vasculatures of control and *Ndufs4^−/−^* mice were very similar (Figure 1F and Appendix A). Finally, the neurovascular system was also examined by standard histological methods. One-month-old *Ndufs4^−/−^* mice and control littermates were euthanized and their brains were cut and stained for the endothelial cell marker CD31. Fluorescence scanning images showed a wide distribution of blood vessels throughout the brain. Size and branching of the blood vessels were very similar between the two genotypes. Confocal microscopy analysis in three different regions (i.e., vestibular nucleus, cortex and olfactory bulb) showed no major differences or structural abnormalities in the brain of *Ndufs4^−/−^* mice compared to controls (Figure 1G). These results showed that no major anatomical alterations of blood vessels are observed in the brain of 1-month-old *Ndufs4^−/−^* mice compared to aged-matched littermates.

### 2.2. Analysis of AAV9-GFP Vector Distribution in Ndufs4^−/−^ and Control Mice

After having confirmed that brain vessels from *Ndufs4^−/−^* mice are comparable to those of wild-type littermates, we wanted to examine whether functional differences existed. To this end, our approach was to test the permeability of the BBB to cargos and molecules of decreasing size ranging from AAV9 vector (25 nm) to Evans blue (960 Da). We first assessed the ability of an AAV9-GFP vector to cross the BBB in adult *Ndufs4^−/−^* and control mice. A previous study using intravenous injection of an AAV9-*Ndufs4* vector in neonates to reinstate gene expression showed that the transgene was expressed in the brain [20]. Whilst AAV vectors injection in neonates represented an attractive method to deliver genes into the brain parenchyma, BBB permeability to AAV vectors in older *Ndufs4^−/−^* mice has never been explored. Hence, we performed intravenous injections of an AAV9-CAG-GFP vector (5.0 *×* 10^13^ vg/kg) in 1-month-old *Ndufs4^+/+^* and *Ndufs4^−/−^* mice. Two weeks later, mice were euthanized and their brains and livers were collected. Livers were harvested as positive controls of successful injections as the AAV9 vector has a good tropism for hepatocytes. A histological analysis of brain sections showed that GFP was expressed in rare cells in the brain of control and *Ndufs4^−/−^* mice (Figure 2A). We specifically examined regions known to exhibit a severe pathology in this mouse model of LS (i.e., olfactory bulb and vestibular nucleus) and found no differences between *Ndufs4^−/−^* mice and control littermates [24]. This result was confirmed by Western blot in which GFP expression was barely detectable in brain lysates from both genotypes (Figure 2C). In contrast, histological and Western blot analyses of the liver showed that GFP was widely expressed in this organ thereby confirming that the weak expression of the transgene in the cerebral parenchyma was not due to unsuccessful injections (Figure 2B,C). Together, these results showed that the AAV9 vector poorly transduced the brain at this dose in 1-month-old mice regardless of their genotype.

### 2.3. Evaluation of the Ndufs4^−/−^ Mice BBB Permeability for Evans Blue and Albumin

Because of the large size of viral vectors (around 25 nm), we could not draw the conclusion that the permeability of the BBB was fully maintained in *Ndufs4^−/−^* mice. Hence, we pursued our exploration and we assessed the extravasation of smaller molecules. First, we used Evans blue, a small dye (960 Da) with a strong affinity for albumin that does not cross the BBB in physiological conditions and that is commonly used to assess vascular leakage in the brain [28]. Evans blue was injected intravenously in 1-month-old *Ndufs4*^+/+^ and *Ndufs4^−/−^* mice. In addition, we performed similar injections in control mice aged 14 to 18 months. These old mice served as a positive control for Evans blue extravasation to the brain. Indeed, it has been demonstrated that BBB permeability is increased in aged mice for instance in the hippocampus [4,29]. To allow for a uniform vascular distribution of the Evans blue and to visualize vascular leaks, mice were euthanized 24 h after the systemic injection (Figure 3A). One brain hemisphere was used to evaluate the penetration of the dye by spectrophotometry, and the other half was processed for microscopic analysis. Quantitative measures of the absorbance from brain samples showed no difference between 1-month old *Ndufs4^−/−^* mice and aged-matched littermates (0.049 ± 0.002 and 0.053 ± 0.003, respectively, *p* > 0.05), while absorbance values were increased significantly in 18-month-old wild-type mice (0.068 ± 0.003, *p* < 0.05) (Figure 3B). In line with these findings, fluorescence scanning microscopy images showed no differences in Evans blue intensity in the brain of *Ndufs4^−/−^* mice compared to control animals, and as previously described [4,29], we could detect Evans blue extravasation in various regions of the brain in old mice (Figure 3C). Livers were also processed for histology as positive controls and showed no differences in Evans blue staining between experimental groups (Figure 3D). Although we used a standard method, this 24 h delay between the injection of the dye and the sacrifice represents a marginal time window with respect to the age of the animals. Experiments using Evans blue are useful to identify prominent leakage but the detection of endogenous plasma proteins is better to detect milder defects and abnormal passage over a longer period of time.

In a second set of experiments, we studied the permeability of the BBB to albumin, a small (66 kDa) native plasma protein that accumulates in the brain during ageing [4,29]. Immunostaining for this protein showed no difference in intensity in 1-month-old *Ndufs4^−/−^* mice compared to aged-matched animals (Figure 4A). In aged animals, albumin was detected in various brain regions including the olfactory bulb, the hippocampus and the vestibular nucleus (Figure 4A). The specificity of the albumin staining was confirmed by the detection of the protein in the organ by which it is produced, namely the liver (Figure 4B). These results were confirmed by Western blot analysis showing the presence of albumin in the brain of aged control mice and in the liver of all animals, while being undetectable in the brain of young mice regardless of their genotype (Figure 4C). Therefore, the BBB permeability to small endogenous proteins and exogenous dyes appeared unaffected in 1-month-old *Ndufs4^−/−^* mice.

## 3. Discussion

Mitochondrial dysfunction occurs during organismal senescence and is observed in a plethora of genetic and sporadic diseases in which increased BBB permeability was also reported. Establishing the direct causative role of mitochondrial dysfunction on the vascular integrity is quite challenging as several pathogenic mechanisms contribute simultaneously to the disease. In this study, we chose to address this question in an in vivo model of primary mitochondrial defect. LS is the most common and one of the most severe mitochondrial diseases. This pediatric condition causes major neurometabolic symptoms that lead to the death of patients within the first decade of life [30]. To date, there is no cure for this fatal disease and LS patients are only offered symptomatic and palliative treatments [31]. Among the ~90 genes associated with LS, the ablation of murine *Ndufs4*, a gene encoding a structural protein of mitochondrial complex I, faithfully replicates cardinal symptoms observed in LS patients [24,25,27]. Hence, *Ndufs4^−/−^* mice represent a relevant model to explore the consequence of mitochondrial dysfunction of BBB integrity. Previous studies in *Ndufs4^−/−^* mice showed that tissues are differentially affected and this correlates with their reliance on the oxidative phosphorylation system [20,24]. Here, we demonstrated that complex I activity is severely impaired in cells constituting the BBB in this mouse model. Structurally, this does not have a major impact on the cerebral vasculature nor on the expression of proteins of the tight and adherens junctions. The absence of anatomical or molecular abnormalities does not predict a normal permeability as one cannot not assess every single component of these junctions. Therefore, we performed functional experiments with molecules of various size ranging from ~25 nm (AAV vectors) to ~1 kDa (Evans blue). Our results showed that the deletion of *Ndufs4* does not affect the permeability to an AAV9-GFP vector injected systemically in 1-month-old mice. These data confirm the rationale of previous studies including ours using methods to bypass the BBB in this mouse model of LS. This included the direct delivery of the viral particles by stereotaxic surgery [19,20,21], the injection during the postnatal period at a time where the BBB is still open [19,20], and the administration of novel AAV capsids with increased ability to cross the BBB [22,23]. Together with the finding that brain-specific deletion of *Ndufs4* replicates pathological features of LS [27], these studies consolidate the idea that gene replacement strategies must target the central nervous system in order to provide a robust therapeutic effect.

Then, we tested the permeability of the BBB to smaller molecules using standard methods. Our data showed that albumin (66 kDa) bound or unbound to Evans blue cannot pass the BBB neither in young *Ndufs4^−/−^* mice nor in control littermates. Experiments based on the systemic injection of Evans blue (960 Da) assess not only the short-term passage of albumin but also that of the dye itself from the unbound fraction [32,33]. Nevertheless, we cannot exclude the possibility that molecules smaller than Evans blue could pass the BBB in this mouse model.

Transporter systems play an important role in BBB permeability, most importantly by regulating the entry of more than 90% of the drugs into the central nervous system. This active transport was not examined here because several studies tested the therapeutic effect of candidate compounds in this mouse model of LS. Thereby, the integrity of selective transports has been indirectly investigated. For instance, Perry and colleagues examined concentration of doxycycline (~444 Da) in the brain of *Ndufs4^−/−^* mice and control littermates and found no difference [17]. These transporters require ATP consumption. It has been previously described that disruption of complex I activity in this mouse model of LS does not cause a reduction in ATP level [24]. This supports the hypothesis that ATP-dependent transport is unaffected in this mouse model. Nevertheless, future pharmacological strategies in this model will have to take into consideration the size and the ability of candidate molecules to cross the BBB.

This study is the first to examine the structural and functional integrity in a genetic mouse model of primary mitochondrial deficiency. In 2015, Doll and colleagues reported that rotenone-induced complex I inhibition disrupts the BBB in in vitro and in vivo models. The discrepancy between this report and our findings can be explained by the difference in methodologies. As nicely described by the authors as “a mitochondrial crisis”, this pharmacological approach triggers a sudden and very acute inhibition of complex I activity that is unlikely to be replicated in transgenic mice or in patients. We hypothesize that the residual activity of complex I in *Ndufs4^−/−^* mice is sufficient to maintain a permeability similar to that in control mice [20,24,25]. This suggests that a major impairment in oxidative phosphorylation in cells composing the BBB does not significantly affect their function and that they may rely mostly on other metabolic sources [34].

With the current set of results, we can only draw the conclusion that the BBB is not structurally or functionally altered in 1-month-old *Ndufs4^−/−^* mice. To broaden this conclusion, it will be essential to carry out similar experiments in other genetic mouse models of LS such as *Surf1^−/−^* mice that present a deficiency in complex IV activity [35]. One step further will be to assess the permeability of the BBB in other models of mitochondrial diseases [36]. Mice with coenzyme Q10 deficiency would be good candidates as Wainwright and colleagues [12] previously reported that the pharmacological depletion of this coenzyme via the inhibition of its biosynthetic pathway in porcine and murine endothelial cell lines leads to an increase in BBB permeability. This would allow us to further ascertain the robustness of approaches combining in vitro models with pharmacological treatments by comparing these results to more relevant in vivo models.

Considering that mitochondrial diseases are a very heterogenous family with a wide spectrum of presentation, the fact that we investigated this question in one of the more severe models of mitochondrial disease does not preclude that milder forms caused by deletion of other genes associated with the oxidative phosphorylation system can disrupt the BBB. Since *Ndufs4* encodes for a structural sub-unit of complex I, it is possible that deficiency in mitochondrial proteins with enzymatic activities may have different consequences on BBB permeability. This stresses the importance of exploring this question in complementary models without assuming that similar results will be found.

We cannot exclude the possibility that the BBB may be compromised in older *Ndufs4^−/−^* mice. One-month-old animals were chosen on the basis of previous gene therapy studies and the fact that it corresponds to a time window at which patients could be treated [22,23]. It has been recently shown that 50-day-old *Ndufs4^−/−^* mice in which brain-resident microglia have been depleted exhibit peripheral macrophages at the site of necrotic lesions [37]. This may reflect a region-specific collapse of the BBB at the end-stage of the disease and the subsequent invasion of circulating immune cells. These findings contribute to the continuous characterization of this highly relevant model of LS that remains a powerful tool to explore disease-related mechanisms and to test therapeutic strategies. Unfortunately, we cannot compare these results to clinical observations as no study has yet explored the permeability of the BBB in children affected with LS.

Altogether, this study represents the first report on the structural and functional integrity of the BBB in a standard model of primary mitochondrial deficiency.

## 4. Materials and Methods

### 4.1. Animals

*Ndufs4* heterozygous mice were obtained from Jackson Laboratories (stock number 27058) and bred to produce *Ndufs4^−^^/−^* offsprings. Mice were genotyped at two weeks of age using the protocol from Jackson Laboratories and the following primers: wild-type forward: AGTCAGCAACATTTTGGCAGT; common: GAGCTTGCCTAGGAGGAGGT; mutant forward: AGGGGACTGGACTAACAGCA. In order to provide warmth and social interaction, *Ndufs4^−/−^* mice were housed with a minimum of one control littermate. Water bottles with long tip and food on the bottom of the cage were provided for each cage containing knockout mice so that access to food and water was not a limiting factor for survival. Mice were observed every other day and euthanized if they showed a 20% loss in maximum body weight, lethargic behavior, or were found prostate or unconscious.

### 4.2. Three-Dimensional Mapping of the Brain Vasculature

One-month-old *Ndufs4^−/−^* mice and age-matched control littermates were anesthetized with isoflurane and received a retro-orbital injection of Evans blue (4% in NaCl 0.9%) (Santa Cruz Biotechnologies, Dallas, TX, USA, sc-203736). The dye was allowed to circulate for at least 5 min as previously described [38]. Mice were then euthanized by an intraperitoneal injection of pentobarbital (Exagon, 100 mg/kg) and the chest was opened to expose the beating heart. An additional 200 μL of Evans blue solution was slowly injected (100 μL/min) via the left ventricle and the needle was kept in place for an additional minute. The brain was then dissected and placed in 4% PFA at 4 °C for 24 h with gentle agitation. After fixation, the brain was washed three times in PBS and stored in PBS with 0.05% sodium azide at 4 °C before proceeding with the clearing procedure. Brains were cleared using the Fast 3D Clear method [39] with minor modifications. Briefly, brains were dehydrated with increasing concentrations of tetrahydrofuran (THF) (Sigma-Aldrich, Saint-Quentin-Fallavier, France) (50 to 90%, diluted in dH_2_O and pH 9.0 adjusted with triethylamine at a series). They were then rehydrated (from 90 to 50% THF solutions) and washed several times with dH_2_O. All steps were performed at 4 °C with gentle rotation. Brains were incubated in the Histodenz-based clearing solution (Sigma-Aldrich) with a refractive index of 1.515 at 37 °C with rotation for several hours.

All brains were imaged using a light sheet microscope (Ultramicroscope II, Miltenyi Biotec, Germany) equipped with an sCMOS camera (Andor Neo) and an MV PLAPO 2XC 0.5 NA objective (Olympus, Rungis, France). Cleared brains were mounted on a sample holder and imaged in type A Cargille immersion oil (Cargille, Cedar Grove, NJ, USA) with a refractive index of 1.518. The cerebral vasculature was visualized by Evans blue fluorescence (using a 561 nm excitation filter and a 620/40 nm emission filter). We maximized the signal-to-noise ratio to avoid saturation by optimizing the excitation power (14%) so that the strongest signal did not exceed the dynamic range of the camera. In the z dimension, we took sectional images in 3 µm steps while using left- and right-sided illumination and sheet width of 40% and NA = 0.156 with dynamic focus (step number:18). The measured resolution was 3.158 µm, 3.158 µm and 3 µm for x, y and z, respectively.

### 4.3. Purification of Brain Vessels

The protocol was performed as previously described [40]. Briefly, mice were euthanized by intra-peritoneal injection of pentobarbital (Exagon, 100 mg/kg) and perfused through the ascending aorta with PBS. Brains were collected in an ice-cold dissection buffer (1% HEPES 1M in HBSS) and cut in 2 mm pieces before being homogenized. The lysate was centrifuged at 2000 g for 10 min at 4 °C and the supernatant was removed. The pellet was resuspended in a purification buffer (1% HEPES, 18% Dextran in HBSS) and vigorously shaken for 1 min before being centrifuged at 4400× *g* for 15 min at 4 °C. The supernatant containing the myelin was discarded and the pellet was resuspended in a conservation buffer (10 mM HEPES, 1% BSA in HBSS). The quality of this purification method was verified by analyzing the expression of the endothelial cell marker CD31 by Western blot as described below (Figure 1A).

### 4.4. Mitochondrial Complex I Activity Assay

Cerebral vessels isolated from *Ndufs4^−/−^* and control mice were used for analysis of mitochondrial complex I (NADH:ubiquinone oxidoreductase) using a colorimetric method (ab109721, Abcam). Mitochondria were purified using the Qproteome mitochondria isolation kit (Qiagen). Oxidation of NADH was assessed by measurement of the absorbance at 450 nm using a PHERAstar FS plate reader (BMG LABTECH, Champigny s/Marne, France).

### 4.5. Evans Blue Assay

One-month-old *Ndufs4^-/-^* mice, aged-matched littermates and 18-month-old *Ndufs4^+/+^* mice were anesthetized with isoflurane (2% for control mice and 0.5% for *Ndufs4^−/−^* mice due to their hypersensitivity to volatile anesthetics) [41,42]. Evans blue was prepared as 4% solution in PBS and was injected as a single bolus dose of 2 mL/kg via the retro-orbital sinus. After 24 h, mice were deeply anesthetized by intra-peritoneal injection of pentobarbital (100 mg/kg) and perfused through the ascending aorta with PBS. Tissues were processed for histological and biochemical analyses.

### 4.6. Analysis of Evans Blue Content by Spectroscopy

The protocol was adapted from a previous study [43]. One brain hemisphere was weighed and incubated in 50% trichloroacetic acid (at 1:4 weight-volume ratio) overnight under agitation to extract Evans blue. Samples were centrifuged for 30 min at 10,000 rpm to pellet the tissue and supernatants were collected in separate tubes. Absorbance of the supernatant was measured at a wavelength of 620 nm using a PHERAstar FS plate reader (BMG LABTECH).

### 4.7. Adeno-Associated Viral Vector Production, Titration and Injection

HEK293T cells grown in DMEM supplemented with 10% FBS and 1% penicillin-streptomycin were triple transfected with pHelper (#240071, Agilent Technologies, Les Ulis, France), pAAV-CAG-GFP (a gift from Edward Boyden, catalog number 37825, Addgene, Watertown, MA, USA), and pAAV2/9n (a gift from James M. Wilson, catalog number 112865, Addgene, Watertown, MA, USA) plasmids using polyethylenimine (Polyscience Inc., #23966, pH 7.0). After 72 h, media and cells were collected and processed separately. Cells were lysed in a hypotonic buffer and nuclei were pelleted by centrifugation. Nuclear pellet was lysed and genomic DNA was degraded by sonication followed by incubation with DNAse (1 UI/µL) and RNase (1 mg/mL). Media was filter in a Stericup 0.22 μm PES membrane and AAV particles were precipitated with polyethylene glycol 8000 and collected by centrifugation. AAV vector particles from media and cells were then purified using discontinuous iodixanol gradient ultracentrifugation. Viral particles were collected using a syringe attached to a 22 G needle to pierce into the centrifugation tube between the 40% and 60% layers. The viral vector solution was then washed in PBS and concentrated using a Millipore Amicon Ultra filter unit (#UFC910008, 100 kD). For the titration, all extra-viral DNA was removed by digestion with DNase I (1 unit/µL). Then, the viral DNA was released by lysis with 2 M NaOH. RT-qPCR was performed on extracted viral DNA and a serial dilution of a viral plasmid containing ITRs as a standard using SsoAdvanced Universal SYBR Green Supermix (Bio-Rad, Hercules, CA, USA), primers against the ITRs (forward: GACCTTTGGTCGCCCGGCCT; reverse: GAGTTGGCCACTCCCTCTCTGC) and the CFX96 Touch Real-Time PCR Detection System and the CFX Maestro Software (version 2.0, Bio-Rad, Hercules, CA, USA). To obtain the final concentration of the viral solution (expressed in vg (viral genome)/mL), the CT values of vg copies were calculated using the standard curve. The titer of the AAV9-CAG-GFP vector was 6.11 × 10^14^ viral genomes (vg)/mL.

### 4.8. AAV Vector Injection

At one month of age, *Ndufs4^−/−^* mice and control littermates were anesthetized with isoflurane as described above. They received an intravenous injection of the AAV9-CAG-GFP vector at a dose of 5.0 × 10^13^ vg/kg (in PBS) through the retro-orbital sinus. Two weeks after this procedure, animals were euthanized for histological and biochemical analyses.

### 4.9. Tissue Staining and Microscopy

For the analysis of Evans blue distribution, one brain hemisphere and one lobe of the liver were post-fixed in 4% paraformaldehyde (in PBS) for 5 days. Organs were then cryoprotected overnight in 30% sucrose (in PBS) before being embedded in OCT (MicronMicrotech) and cut on a cryostat (CryoStar NX50, Thermo Fisher Scientific, Villebon-sur-Yvette, France). Livers and brains were cut as 14 μm- and 35 μm-thick sections, respectively. Nuclei were stained with DAPI (5 μg/mL in PBS) (Invitrogen) for 5 min before being rinsed five times with PBS. Sections were coverslipped using the Dako fluorescent mounting medium (DAKO, Agilent, Les Ulis, France).

For other histological procedures, mice were deeply anesthetized by intra-peritoneal injection of pentobarbital (100 mg/kg) and transcardially perfused through the ascending aorta with PBS and then with 4% paraformaldehyde diluted in PBS. Tissues were dissected and post-fixed for 24 h at 4 °C in 4% paraformaldehyde. Tissues were cryoprotected overnight in 30% sucrose before being embedded in OCT and cut on a cryostat (Thermo Fisher Scientific CryoStar NX50).

Histological staining was performed as previously described [22]. Sections were rinsed in PBS before being incubated for 1 h in 0.1 M PBS containing 10% normal goat serum and 0.25% Triton X-100. Tissues were then incubated overnight at room temperature in the same solution containing one of the following primary antibodies: anti-GFP (chicken, 1:2000, Abcam, #ab13970), anti-albumin (chicken, 1:1000, Abcam, #ab106582), anti-CD31 (goat, 1:500, R&D Systems, AF3628, Noyal Châtillon sur Seiche, France). Sections were then washed three times in PBS and incubated with secondary antibodies coupled to Alexa 488 or Alexa 568 (1:400, Molecular Probes, Invitrogen) diluted in the blocking solution for 2 h at room temperature. They were rinsed again and nuclei were stained with DAPI (5 μg/mL in PBS) (Invitrogen) for 5 min before being rinsed five times in PBS. Sections were coverslipped using the Dako mounting medium (Agilent, Les Ulis, France). Images were acquired on a LSM 710 confocal microscope (Zeiss, Reil-Malmaison, France) or an AxioScan.Z1 slide scanner (Zeiss).

### 4.10. Western Blotting

Tissues were homogenized in RIPA buffer (Sigma, Saint-Quentin-Fallavier, France) and Western blot procedure was performed as previously described [22]. Protein concentration was measured using the Bio-Rad DC protein assay. Ten micrograms of protein were boiled at 95 °C for 5 min in Laemmli buffer (Bio-Rad, Hercules, CA, USA) containing 5% β-mercaptoethanol (50 mM), separated on a SDS-PAGE gel and then electrotransferred (100 V, 1 h) on a PVDF membrane (pore size: 0.45 µM) (Millipore, Fontenay sous Bois, France). After blocking for 1 h in Tris-buffered saline with 0.1% Tween-20 (TBS-T) and 5% non-fat dry milk (Sigma-Aldrich), membranes were incubated overnight at 4 °C with one of the following primary antibodies: GFP (chicken, 1:2000, Abcam, #ab290), calnexin (rabbit, 1:5000, Enzo Life Science, ADI-SPA-860), GAPDH (rabbit, 1:3000, Cell Signaling, Boston, MA, USA #2118), CD31 (goat, 1:500, R&D Systems, AF3628), occludin (rabbit, 1:10,000, Proteintech, #27260-1-AP), GFAP (goat, 1:3000, Santa Cruz Biotechnologies, sc-6170), claudin5 (mouse, 1:1000, Thermofisher, #35-2500) or albumin (chicken, 1:1000, Abcam, #ab106582). After washing for 30 min in TBS-T, membranes were incubated for 1 h at room temperature with an HRP-conjugated secondary antibody (1:3000, Jackson Immunoresearch, West Grove, PA, USA). Proteins were revealed with the Clarity kit (Bio-Rad) or the Immobilon Western kit (Millipore). Luminescence signal was detected using the ChemiDoc MP (Bio-Rad) and images were analyzed with ImageJ software (version 1.52).

### 4.11. Statistical Analysis

Statistical analysis was conducted using the GraphPad Prism software (version 9). One-way ANOVA test with Tukey’s multiple comparison test or Student’s t-test were performed to analyze the difference between experimental groups. The data were collected and processed in a randomized and blinded manner. No statistical methods were used to predetermine sample size, but our sample sizes are similar to those generally employed in the field. All values are presented as mean ± standard error of the mean (SEM). Statistical significance was set at *p* < 0.05.

## Figures and Tables

**Figure 1 ijms-25-04828-f001:**
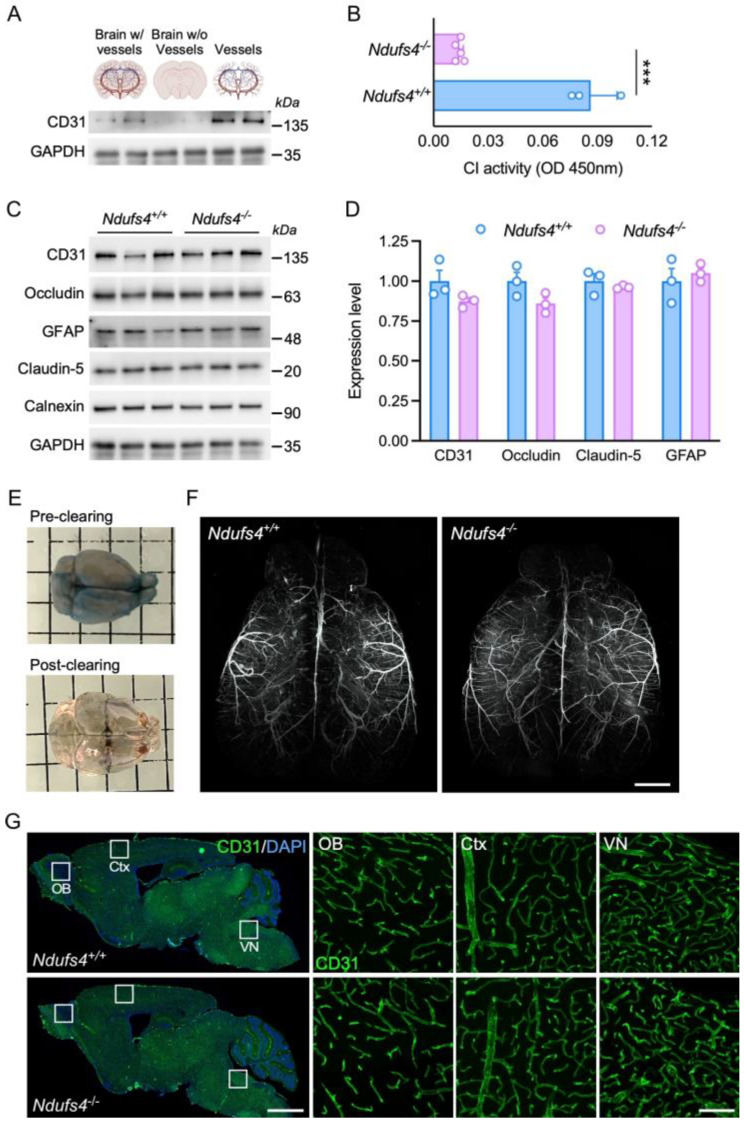
Biochemical and anatomical analysis of cerebral blood vessels. (**A**) Western blots showing the expression of CD31 and GAPDH in crude brain tissue, the discarded fraction from the purification and the purified fraction containing cerebral blood vessels. (**B**) Mitochondrial complex I (NADH:ubiquinone oxidoreductase) activity assay from cerebral blood vessels of 1-month-old control (n = 3) and *Ndufs4^−/−^* mice (n = 5). *** *p* < 0.001 (Student’s *t*-test). (**C**) Western blots showing the expression of CD31, Occludin, GFAP and Claudin-5 in 1-month-old *Ndufs4^+/+^* and *Ndufs4^−/−^.* Calnexin and GAPDH were used as loading controls. (**D**) Quantification of the Western blots (n = 3 per group, Student’s *t*-test). (**E**) Images showing a mouse brain before and after the Fast3D clearing protocol. (**F**) Three-dimensional reconstruction of light sheet fluorescence microscopy images of brains from 1-month-old control and *Ndufs4^−/−^* mice injected with Evans blue. Scale bar: 1.5 mm. (**G**) Fluorescence scanning microscopy images showing sagittal brain sections from 1-month-old control and *Ndufs4^−/−^* mice stained for CD31 (green) and DAPI (blue). Scale bar: 1.5 mm. Inserts show confocal microscopy images of specific regions marked with squares. Scale bar: 100 μm.

**Figure 2 ijms-25-04828-f002:**
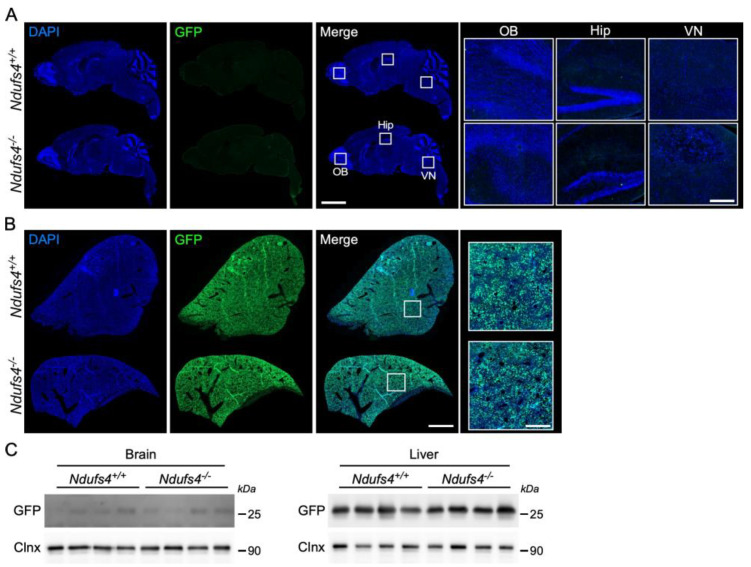
GFP expression in the brain and liver two weeks after intravenous injection of an AAV9-GFP vector in 1-month-old mice. (**A**) Scanning microscope images of sagittal brain sections from control and *Ndufs4^−/−^* mice showing the expression of GFP (green). Sections are counterstained with DAPI (blue). Scale bar: 2 mm. Right panel shows an enlargement from the olfactory bulb (OB), the hippocampus (Hip) and the vestibular nucleus (VN). Scale bar: 200 μm. (**B**) Scanning microscope image of a representative liver section from control and *Ndufs4^−/−^* mice showing the expression of GFP (green) and stained with DAPI (blue). Scale bar: 2 mm. Right panel shows an enlargement of a representative area of the section. Scale bar: 250 μm. (**C**) Western blots showing the expression of GFP and calnexin in the brain and liver of control and *Ndufs4^−/−^* mice two weeks after injection of an AAV9-GFP vector.

**Figure 3 ijms-25-04828-f003:**
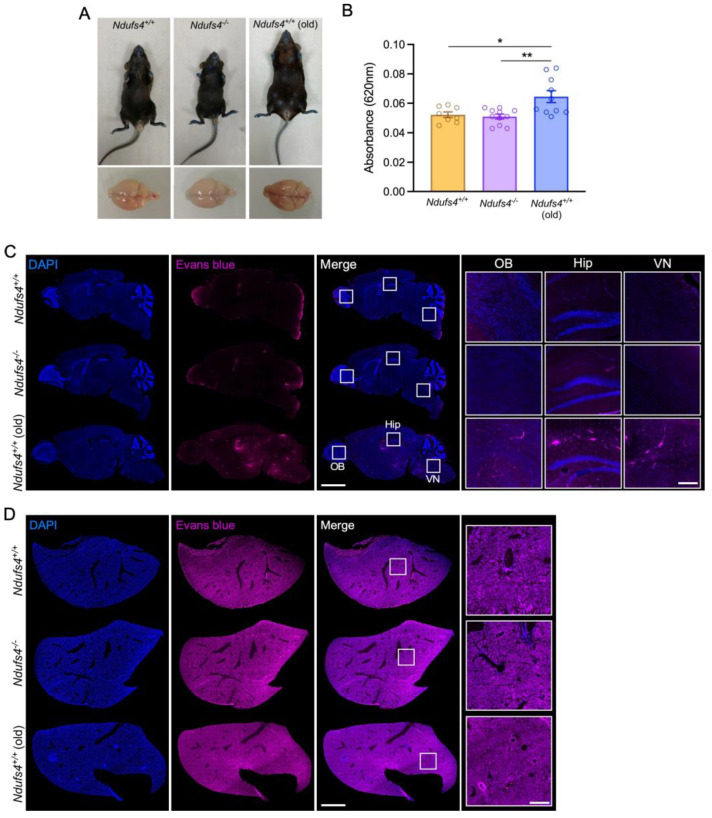
Analysis of Evans blue extravasation. (**A**) Representative pictures of 1-month-old control mice, *Ndufs4^−/−^* mice and 18-month-old control mice (upper panel) and their brains (lower panel) 24 h after an intravenous injection of Evans blue. (**B**) Absorbance measurement at 620 nm of Evans blue extracted from 1-month-old *Ndufs4^+/+^* mice (n = 8), *Ndufs4^−/−^* mice (n = 11) and 18-month-old control brains (n = 10). * *p* < 0.05; ** *p* < 0.01 (one-way ANOVA). (**C**) Scanning microscope image of a sagittal brain section from 1-month-old control, *Ndufs4^−/−^* mice and 18-month-old control mice imaged at 650 nm to detect Evans blue fluorescence (pink) 24 h after intravenous injection of the dye. Scale bar: 2 mm. Right panel shows an enlargement of the olfactory bulb (OB), the hippocampus (Hip) and the vestibular nucleus (VN). Scale bar: 200 μm. (**D**) Scanning microscope image of liver sections from 1-month-old control, *Ndufs4^−/−^* mice and aged control mice stained with DAPI (blue) and imaged to detect Evans blue fluorescence (pink) 24 h after an intravenous injection of the dye. Scale bar: 2 mm. Right panel shows an enlargement of a representative area of the section. Scale bar: 250 μm.

**Figure 4 ijms-25-04828-f004:**
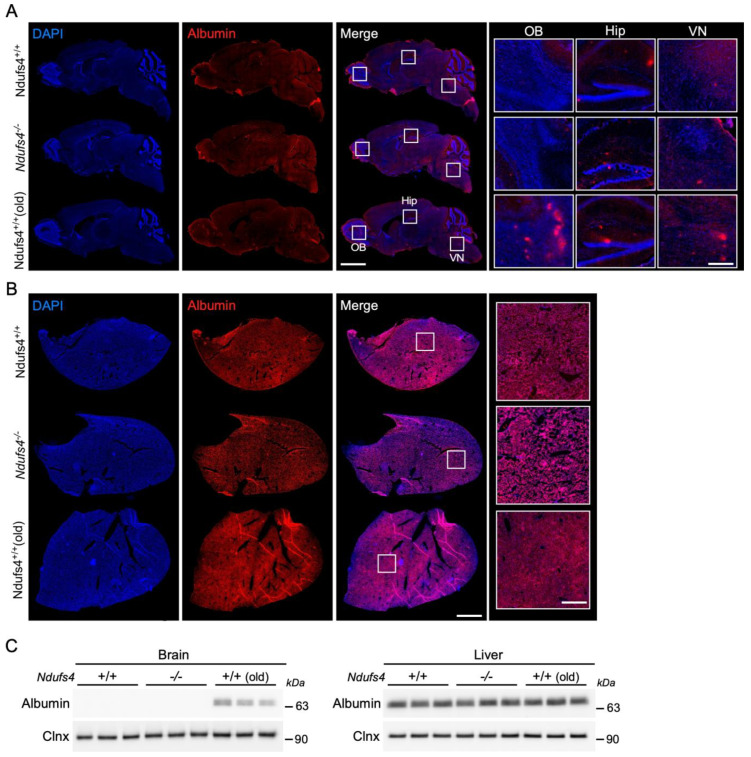
Analysis of albumin extravasation (**A**) Scanning microscope images of sagittal brain sections from 1-month-old control, *Ndufs4^−/−^* mice and old control mice brain showing the expression of albumin (red). Sections are stained with DAPI (blue). Scale bar: 2 mm. Right panel shows an enlargement of the olfactory bulb (OB), the hippocampus (Hip) and vestibular nucleus (VN). Scale bar: 200 μm. (**B**) Scanning microscope images of liver sections from 1-month-old control, *Ndufs4^−/−^* mice and 18-month-old control mice showing the expression of albumin (red). Sections are stained with DAPI (blue). Scale bar: 2 mm. Right panel shows an enlargement of a representative area of the section. Scale bar: 250 μm. (**C**) Western blots showing the expression level of albumin and calnexin in the brain and liver of 1-month-old control and *Ndufs4^−/−^* mice and 18-month-old control mice.

## Data Availability

All the data and reagents (if not commercially available) that support the findings of this study are available from the corresponding author upon reasonable request.

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
