# Peer review of "The Blood–Brain Barrier Is Unaffected in the Ndufs4−/− Mouse Model of Leigh Syndrome"

_ijms, 2024, doi:10.3390/ijms25094828_

Round 1
Reviewer 1 Report
Comments and Suggestions for Authors
The present study demonstrates the structural and partial functional integrity of the blood-brain barrier (BBB) is preserved in the Ndufs4-/- mouse model of Leigh syndrome, and its permeability to specific molecules of different sizes is examined. Research work has good scientific significance.
However, I have some main inquiries regarding the design in evaluation of BBB function, and have raised the following questions:
Major:
The BBB is not only a static physical bio-membrane barrier, but also an active and selective transporter barrier. There are numerous transporter families on the BBB, such as MDR transporters, MRP transporters, BCRP transporters, OATPs, and others. They play an important role in limiting the permeability of 95-98% of drugs entering the central nervous system through the BBB.
Moreover, the uptake or efflux process participated by the transporters is an energy (ATP) consuming process, which is closely related to mitochondria.
Therefore, why has the important role of transporter systems been overlooked in the functional evaluation of BBB?
Major:
In this study, the author team selected Evans blue and albumin as markers for evaluating the permeability of BBB. However, the molecular weight of Evans Blue is around 960 Da. It cannot be used as a single representative of small molecule. The classic markers for evaluating BBB permeability include Tenoxicam, Propranolol, Inulin, Mannitol, Caffeine and Retinol, etc. Furthermore, it is necessary to calculate the permeability coefficients (Papp) for comparison with literature and statistical analysis.
Major:
The article title is covering a wide range, is too broad. In fact, it does not include conventional small molecule drugs.
In addition, the following minor issues need to be addressed to further improve the manuscript:
Minor: Are the contents of 2.2 and 2.3 same in the results section? Please check.
Minor: Since the BBB permeability is increased in aged mice (Line 207), why was the AAV9-GFP vector distribution not analyzed in aged mice in the section 2.2?
Minor: In Figure 1A, the first letter of “vessels” should be capitalized.
Minor: In the title of the article, "mouse" should be specifically written as "Ndufs4-/- mouse".

There are some minor writing mistakes.
Reviewer 2 Report
Comments and Suggestions for Authors
Robin Reynaud-Dulaurier et al. conducted experiments in vivo to explore whether mitochondrial disease will affect the blood-brain barrier in Ndufs4-/- mouse model. The introduction gives the motivation of this study and sufficient background. The conclusion is well supported by the results. I am very impressed by the pictures and the techniques in the animal operation. However, the presentation of the data has a lot of confusions. After careful consideration, I recommend accepting this paper for publishing in International Journal of Molecular Sciences with major corrections. Comments and questions are as follow:
1. No period presenting at the end of some paragraphs (e.g. line 348, 377, 403, 410).
2. In line 352, the data is presented as 0,9% which is supposed to be 0.9%. This type of typos can be found all over the paper.
3. There are several units error for my understanding. For example, line 355 and 356, 200 mL Evans blues injection is impossible for a mouse. And I do not think 100 mL/min injection is a slow one; line 420, I cannot read the unit after 0.22.
4. Line 363, H2O should be H2O.
5. Line 435 and 439, I cannot understand the numbers of 6,11.1014 vg/ml and 5.1013 vg/kg.
6. Line 446, I do not believe the frozen tissue sections were prepared 14 mm and 35 mm thick.
7. Scale bar for the pictures should not be in mm.
Comments on the Quality of English LanguageThe mainly expression has no problem. But the data presentation with numbers and units need to be carefully revised.
Reviewer 3 Report
Comments and Suggestions for Authors
The purpose of the study is interesting for the scientific and clinical community. However, some points are relevant for further clarification and corrections.
Considering the writing of the paper, I think a good review is relevant to correct typos, formatting and grammatical improvements
Results: The description of the results with a brief discussion of the purpose of the analysis is interesting, but as the information develops the reader ends up getting lost in the results found at each stage of the experiment. I suggest a brief closing of the topics with a summary to make it clear what the study found in that analysis purpose. This was even done at times.
The graph in Figure 3 could be a boxplot instead of a histogram to better analyze the dispersion of the data.
Decimals must be separated by periods and not commas.
The discussion could be a little more direct, since parts were already described in the introduction and there was a lack of a clearer and more objective beginning, showing what the study contributed again in this context. The manuscript became a little tiring with so much justification and little clarity and objectivity. There was also a lack of a summary of the study’s limitations. A conclusive end to the study is necessary.
The methods were described satisfactorily
Comments on the Quality of English LanguageConsidering the writing of the paper, I think a good review is relevant to correct typos, formatting and grammatical improvements
